# Assessing forest carbon offset additionality with dynamic baselines and uncertainty quantification

### Noah Golmant
noah@pachama.com
Pachama, Inc.

### Martha Morrissey
martha@pachama.com
Pachama, Inc.

### Carlos Silva
carlos@pachama.com
Pachama, Inc.

### Felix Dorrek
felix@pachama.com
Pachama, Inc.

### Rachel Engstrand
rachel@pachama.com
Pachama, Inc.

## ABSTRACT

Forest carbon projects, like those reducing emissions from deforestation and degradation (REDD+), can help mitigate climate change by sequestering carbon. Their effectiveness is measured against a "baseline" scenario, which predicts emissions if no project intervention occurred. To maintain the integrity of carbon credits, it's crucial to have accurate baseline emission reduction models with well-characterized uncertainty. However, recent scrutiny has raised concerns about the accuracy of emission reduction claims made by REDD+ projects. These projects rely on *ex-ante* predictions of future deforestation risk with no standard approach to quantify uncertainty. *Ex-post* ("dynamic") baselines could reduce this model uncertainty, but they also lack a standardized framework for uncertainty. We introduce a dynamic baseline model based on remote sensing data and nearest neighbors matching and apply a novel uncertainty quantification framework to assess the accuracy of this model. Applying our approach to seven REDD+ case study projects in Brazil, we found several instances of consistent over or under-estimation of emissions reductions, suggesting potential inaccuracies in current carbon offset measurements. Our findings highlight the importance of our dynamic baseline and uncertainty quantification in enhancing the effectiveness of REDD+ and similar forest carbon projects.

## CCS CONCEPTS

• **Computing methodologies → Machine learning**; • **Applied computing → Environmental sciences**.

## KEYWORDS

climate modeling, nearest neighbors, uncertainty quantification, counterfactuals

**ACM Reference Format:**
Noah Golmant, Martha Morrissey, Carlos Silva, Felix Dorrek, and Rachel Engstrand. 2023. Assessing forest carbon offset additionality with dynamic baselines and uncertainty quantification. In *Proceedings of KDD FragileEarth*

*KDD FragileEarth Workshop '23, August 7, 2023, Long Beach, CA*
© 2023 Association for Computing Machinery.
ACM ISBN 978-1-4503-XXXX-X/18/06...$15.00
https://doi.org/XXXXXXX.XXXXXXX

*Workshop '23*. ACM, New York, NY, USA, 7 pages. https://doi.org/XXXXXXX.XXXXXXX

## 1 INTRODUCTION

The market for forest carbon credits has the potential to make a meaningful contribution to climate mitigation [5]. However, recent analyses have called into question the accuracy of the baselines used to compute project emissions reductions [6, 19, 20]. A baseline represents business-as-usual outcomes without a forest carbon project. The number of credits issued to a landowner annually is the difference between project and baseline carbon emissions. For REDD+ (Reducing Emissions from Deforestation and forest Degradation) projects, the number of credits is determined by reductions in the rate of forest loss in the project area compared to the baseline.

Advances in remote sensing- and machine learning-based approaches to forest cover change detection have enabled more accurate estimates of project carbon emissions [7, 13, 14]. However, there is no established framework to assess the accuracy of REDD+ project baselines [15]. Accurate baselines with well-quantified uncertainty are necessary to ensure that one credit represents at least one tonne of carbon dioxide emissions avoided or removed from the atmosphere. Inaccurate baselines have contributed to several well-publicized instances of over-crediting in which the number of credits issued to a project exceeds the emissions reductions achieved by the project activities [19, 20].

In this work, we propose a pixel-level nearest neighbors-based approach to algorithmically select a baseline with remote sensing data. We also implement an uncertainty quantification framework suitable for validating arbitrary baseline models. We demonstrate the effectiveness of this approach by using it to assess the additionality of seven REDD+ projects in Brazil. This approach provides evidence that it is possible to compute accurate estimates of baseline emissions reductions with well-characterized uncertainty.

## 2 RELATED WORK

### 2.1 Current methodologies

Existing methodologies to compute baselines for REDD+ projects face a number of challenges with respect to baseline accuracy and uncertainty quantification. In Verra's VM0015 methodology for avoided unplanned deforestation, a project developer constructs a baseline by selecting a reference region deemed by an expert to be similar to a project. They then train a deforestation risk model on historical remote sensing data. The project baseline is determined by running inference with this model to predict future deforestation

       

risk in the project area [11, 15]. In other cases, simpler modeling approaches may be used, like constructing univariate linear or stepwise regression models of historical deforestation rates, or assuming the existence of a profit-maximizing agent who would deforest 100% of the property over a short time horizon [15, 16].

There are several issues with these approaches. One problem is that it is impossible to quantify the model (epistemic) uncertainty of the baseline because of the manual steps involved in the modeling process. To assess model uncertainty, the modeling approach must be algorithmic, and the validation process must be able to ensure that the model accurately captures the baseline scenario in areas similar to the project, but where no intervention occurs. Another issue is that these approaches are all *ex-ante*, i.e. they predict future deforestation rates in the vicinity of the project. These rates are difficult to predict and highly sensitive to geopolitical factors—for example, policies enacted in Brazil by former President Bolsonaro contributed to significant increases in deforestation across the country [1]. A model should be robust to the distributional shifts imposed by unpredictable changes in regional deforestation rates.

## 2.2 Dynamic baselines

Recently, there have been efforts to develop *ex-post* counterfactual baseline models to independently assess the additionality of forest carbon projects. [19] introduced the use of the Synthetic Control Method to construct deforestation counterfactuals. They selected synthetic controls from "donor pools" of properties based on accessibility and biophysical characteristics. The project baseline is equal to a weighted combination of the observed forest loss rates in these properties, where the weights are the solution to a bilevel optimization problem. They construct confidence intervals by applying SCM to the synthetic controls, which provides samples of the residual distribution of the model. They found little to no additionality across 12 REDD+ projects in Brazil.

[20] extended this work to analyze 26 REDD+ projects across Peru, Colombia, and several African countries using a Generalized Synthetic Control (GSC) method [21]. They extended the work of [19] across multiple countries by: using a global forest cover change dataset to estimate forest loss [7]; replacing property-based synthetic controls with uniformly sampled circles across the jurisdiction; and constraining the circles to have similar deforestation pressure to the project, as measured by historical forest loss in a 10 km buffer around the boundary.

[6] constructed *ex-post* baselines with a pixel-based nearest neighbor selection approach to analyze 40 REDD+ projects across 9 countries. They matched each pixel in the project to a pixel in a search region using a set of standardized matching covariates. They restricted their analysis to moist tropical rainforests, where they leveraged [14] to estimate both deforestation and degradation activity contributing to forest loss. Due to the high computational complexity of exhaustive k-NN search, they relied on a sub-sampling approach to select nearest neighbor pixels in a search region. They quantified aleatoric uncertainty with non-parametric pixel-level bootstrapping to construct confidence intervals in their baseline estimates.

All of these methods are examples of *dynamic baselines*. A dynamic baseline does not attempt to predict future deforestation rates in the project. Instead, it relies on observing forest cover change in control areas. These control areas are algorithmically selected to be similar to the project with respect to the expected forest loss in the absence of an intervention.

Our approach is novel in several respects. First, we perform exhaustive k-NN searches to eliminate the need for sampling design in [6], matching each project pixel to a nearest neighbor across a large search area around the project. By leveraging parallel computing and vertically scaled compute nodes, we were able to substantially increase the search area compared to prior work to find more accurate nearest neighbor matches. Second, we use remote sensing data sources for all matching covariates, allowing for improved spatiotemporal fidelity in matching. This also reduces the possibility of temporal leakage in our data, e.g. with the introduction of roads developed after the project start date. Third, we introduce an uncertainty quantification approach that is generic enough to characterize model uncertainty of both *ex-ante* and *ex-post* baseline models, allowing for a more standardized validation approach to compare models.

## 3 METHODS

### 3.1 Control area selection: pixel-level matching

We use a k-nearest neighbor (k-NN) algorithm to match the carbon project to a control area within a search region (Figure 1). Each pixel is represented by a feature vector consisting of an array of attributes derived from satellite observations. We match each individual pixel within the project to its nearest neighbor (i.e. the search region pixel with the minimum Euclidean distance in feature space). Matching features are currently weighted equally. The search region is defined as a 100 km buffer around a project restricted to national boundaries.

All remote sensing features are resampled to 100 meter resolution. The control area baselines presented here use the following set of matching features: slope and elevation derived from the SRTM product [10], annual composites of sub-pixel tree cover from the MODIS MOD44B product [2]; annual composites of the MODIS MOD17 gross primary productivity product [12]; distance to recent deforestation as detected by the Global Forest Change product (GFC) [7]; and distances to water and pasture classified by the MapBiomas Collection 7 product [13].

We apply masking to ensure that pixels do not match to existing carbon projects or non-forested areas like water and roads [9, 13]. Additionally, we ensure that the control area pixels lie in regions with the same protected land status as the project [8].

To construct a baseline from a control area, we observe annual forest loss identified by GFC within the control area [7]. To estimate the baseline annual carbon dioxide emissions, we scale the baseline forest loss rate by the total project forest carbon stock estimated by the project developer.

### 3.2 Representative sampling of placebos for uncertainty quantification

Algorithmic baseline approaches enable validation against independent observations using *placebos*. Placebos are randomly selected forested areas without a carbon project. Since there is no carbon

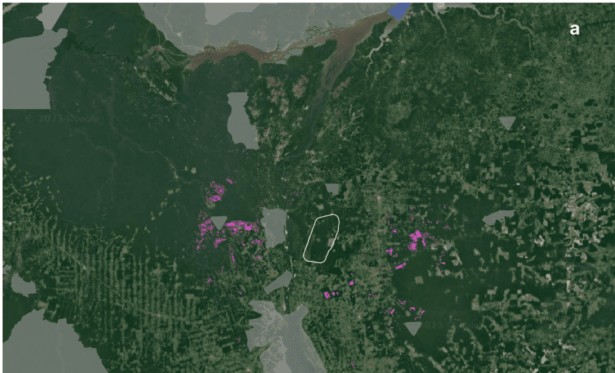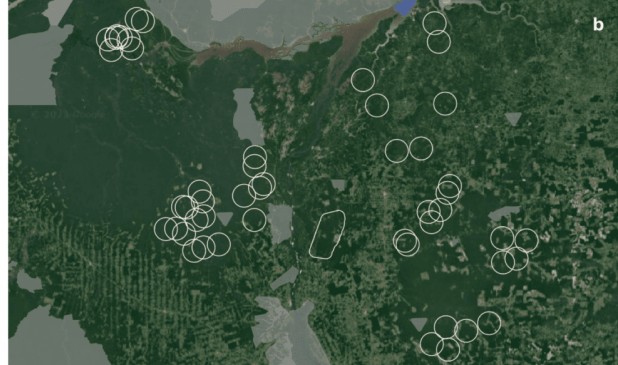

**Figure 1: Left: map of sample anonymized carbon project (white) and its k-NN-selected control area (pink). Right: Map of anonymized carbon project (white) and circular placebo projects (white) used to compute baseline uncertainty. Shaded white areas are existing protected areas, which we exclude when selecting placebos and control areas.**

project, the forest loss predicted by a baseline model should match the observed forest loss in the placebo. The residual error term can be defined as the difference between the predicted baseline and the observed placebo forest loss.

If the placebos are representative of the project, then the residuals should provide an unbiased sample of the error distribution of the baseline model. This enables us to estimate standard regression metrics, such as RMSE, bias, or quantile standard error. Additionally, we can construct a predictive distribution around the baseline by applying a residual resampling technique. Let $\hat{y}_{t,i}$ and $y_{t,i}$ denote the baseline and observed forest loss in hectares for year $t$ and placebo $i$. Let $\hat{y}_t$ denote the baseline forest loss for year $t$ in the project. Then a sample of the predictive distribution, $\hat{Y}_{t,i}$, can be constructed by scaling the residuals by the predicted forest loss:

$$\hat{Y}_{i,t} = \hat{y}_t + \hat{y}_t \frac{y_{t,i} - \hat{y}_{t,i}}{\hat{y}_{t,i}} = \hat{y}_t \frac{y_{t,i}}{\hat{y}_{t,i}} \quad (1)$$

We can also compute cumulative versions of these statistics. Let $\hat{y}_i = \sum_t \hat{y}_{t,i}$ and $y_i = \sum_t y_{t,i}$ denote the cumulative baseline and observed placebo forest loss from project start to present day. Let $\hat{y}$ denote the baseline forest loss for the project. Then each residual results in a sample $\hat{Y}_i$ of the cumulative baseline forest loss distribution:

$$\hat{Y}_i = \hat{y} \frac{y_i}{\hat{y}_i} \quad (2)$$

A central estimate of this distribution, like the median, can be used to assess the additionality of a project. To determine if a baseline independently proposed by a project developer results in over-crediting relative to the dynamic baseline, we can measure where the project developer baseline falls relative to the median.

To ensure that the placebos are representative of the project, we uniformly sample circular boundaries of area equal to the project and apply several filtering criteria. First, we eliminate boundaries that contain carbon projects or do not match the project's protected land status [8]. Second, we compare zonal statistics of several remote sensing attributes to ensure that the mean of each attribute in the placebo lies within one standard deviation of the mean in

the project area. Currently, we consider MODIS MOD44B percent tree cover and distance to recent deforestation [2, 7]. We sample placebos within a 300 km buffer surrounding the project, restricted along national boundaries. In our experiments, we have found this buffer size sufficient to find matching placebos with a sufficient range of observed deforestation rates. Because an exhaustive k-NN approach has high computational and memory demands to run even for a single project, we leveraged distributed computing to horizontally scale computation across the placebos.

## 4 EXPERIMENTS

To assess the behavior of our baseline model and uncertainty quantification framework, we analyzed seven REDD+ projects in the Brazilian Amazon. These projects are presented as case studies. A larger-scale analysis with reduced model uncertainty would be necessary to make accurate judgments about the overall prevalence of over/under-crediting in the global REDD+ voluntary carbon market.

### 4.1 Uncertainty quantification

To quantify model performance, we estimated relative root mean squared error and relative bias using the placebos.

To select the number of placebos, we applied a bootstrapping approach to determine whether the placebos provided a convergent estimate of several metrics — relative RMSE, relative bias, and the 90th percentile of observed deforestation in the placebos. We observed convergence in these metrics after roughly 10 placebos. We increased the sample count to 50 to ensure a wide enough range of observed and baseline forest loss values for residual estimation.

In Table 1 we show uncertainty metrics for the seven REDD+ projects. The range varies significantly on a project-by-project basis, from a minimum relative RMSE of 12% to a maximum of 335%. In Figure 4, we show scatter plots of the placebo baseline versus observed forest loss in each of the placebos for each of the years. We can see that baseline performance varies significantly by project, year, and observed placebo forest loss. Several projects have a tight correspondence across all years, but for other projects,

| Project | Relative RMSE | Relative Bias |
|---------|---------------|---------------|
| Project 1 | 37% | -4% |
| Project 2 | 335% | 149% |
| Project 3 | 99% | 22% |
| Project 4 | 63% | 4% |
| Project 5 | 12% | -5% |
| Project 6 | 63% | -7% |
| Project 7 | 52% | 17% |
| **Average** | **94%** | **25%** |

**Table 1: Uncertainty metrics for each of the anonymized REDD+ projects.**

we see near-uniform predictions for a particular year (as in Project 1), or a wide range of predicted baseline values in placebos with near-zero deforestation, as in Project 3.

The high variance in accuracy between projects suggests that regional, project-agnostic validation of model uncertainty may be inappropriate when assessing the additionality of specific projects. Additionally, model-specific uncertainty quantification approaches, like the donor pool resampling or pixel-level bootstrapping approaches employed in [6, 19], may result in significant under-estimation of model uncertainty.

We have also observed that when uncertainty is too high, as in the case of Project 2, the residual resampling approach may fail to produce a meaningful predictive distribution. Modifications to this uncertainty framework, such as restricting the residual resampling to only include placebos with a similar baseline estimate to the project, may improve performance.

## 4.2 Matching quality

To determine whether the k-NN model yields control areas that are similar to the project, we plotted histograms of the distribution of matching covariates in the project and control area. Figure 5 shows the overlap for an example project. This is similar to previous work, which quantitatively assesses the quality of the dynamic baseline by computing e.g. the standardized difference of means between the matching covariates in the project and control [19, 20].

Although high overlap indicates that k-NN is able to find pixels in the search area that are similar to the project pixels, we have found that they are not necessarily predictive of the uncertainty of the baseline model. For example, historical deforestation *within* the project is not a meaningful covariate, because the project is undisturbed at project start by definition, and this covariate does not capture encroaching deforestation activity outside of the project. The matching quality only correlates with baseline uncertainty if the covariate is predictive of future deforestation. Even then, modeling confounders like feature transformations or the choice of distance metric may alter the strength of this statistical relationship.

Because of this, we judge that an uncertainty framework like the placebo approach is ultimately more important for assessing the quality of the baseline model than other descriptive statistics or diagnostic figures like differences in covariate means.

## 4.3 Assessing the additionality of existing REDD+ projects

To determine whether an existing REDD+ project is significantly over-credited, we compare our dynamic baseline results with the baseline emissions reduction estimates supplied by project developers. By leveraging the baseline distributions we estimate using placebos, we can compare credit estimates both annually and cumulatively.

In Figure 2 we show the time series of annual baseline estimates with confidence intervals for the REDD+ projects. We can see that in some cases, like Project 5, the degree of over/under-crediting varies on an annual basis. For several projects, such as Projects 4 and 7, our results indicate substantial over-crediting. Project 1 is an example where distributional shifts, i.e. accelerating regional deforestation rates, resulted in under-crediting under the project developer's *ex-ante* baseline.

Figure 3 shows the histograms of the cumulative dynamic baseline emissions estimates. In several cases, such as that of Project 4, we can clearly identify substantial over-crediting. In other cases, the conclusions are less obvious, and the model uncertainty contributes to wider tails in the distribution.

## 5 CONCLUSION

In this work, we presented a novel control area-based approach to assess the additionality of forest carbon offset projects with remote sensing data and a generic baseline uncertainty quantification framework. We demonstrated its effectiveness on case studies of Brazilian REDD+ projects, where for several projects we can confidently assert a high degree of over/under-crediting relative to a project developer baseline. Next steps for this work include:

**Improved uncertainty quantification**: Ensuring placebo representativeness remains a challenge. We have seen that even with our current filtering criteria, several projects include placebos that are not similar to the project with respect to properties like the predicted baseline. Placebos with near-zero observed deforestation may skew uncertainty metrics like RMSE or bias, as well as the variance of the predictive distribution. Addressing this may enable us to remove the scaling factor used to account for heteroskedasticity in the residual resampling approach.

**Reducing uncertainty**: Several projects display such high uncertainty that the baseline cannot be used to reliably assess project additionality. Reducing uncertainty is necessary to (a) establish baselines for *ex-post* crediting of new projects and (b) improve the veracity of over-crediting assessments in the market. Some examples of methods to reduce uncertainty include: incorporating additional remote sensing-based predictors of deforestation; introducing learned feature representations, e.g. through semi-supervised approaches like the Vision Transformer [3]; or simpler modifications like non-linear feature transformations or a non-Euclidean metric like the Mahalanobis distance.

**Geographic expansion**: The REDD+ projects analyzed in this work are located in the Brazilian Amazon. Expanding to other countries and forest types requires incorporating new data sources. For example, our land use-based features are derived from MapBiomas, which is only available in Brazil [13]. The accuracy of forest loss detection datasets like GFC also varies between moist tropical and

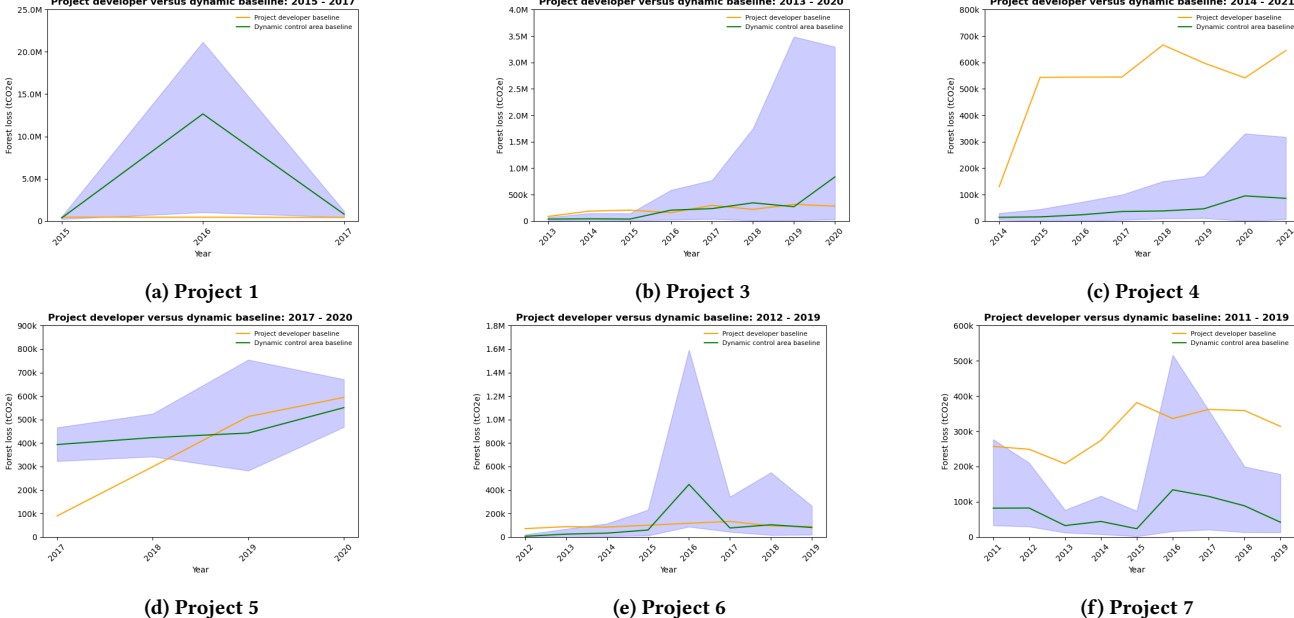

**Figure 2: Baseline time series (green) with P10/P90 confidence intervals for the selected projects (blue), plotted against the project developer baseline (orange). Project 2 is excluded since there is only one year of verified project developer baseline data.**

dry forests [7]. Regions with high cloud cover may require forest loss detection models based on synthetic aperture radar rather than optical imagery. Naively applying the current model to new regions without accounting for these factors may result in high model uncertainty or significant underestimates of forest loss.

**Market implications**: This work focused on assessing the additionality of REDD+ projects, where projects are credited based on reductions in forest loss. A similar framework can be applied to construct *ex-post* baselines and quantify uncertainty for other forest carbon project types. Verra recently approved a dynamic baseline approach for Improved Forest Management (IFM) projects [17], and a similar approach is under development for Afforestation, Reforestation, and Revegetation (ARR) projects [18]. However, REDD+ projects still rely on *ex-ante* baselines. There is also significant room for project developer discretion in baseline construction across these methodologies, and they still lack a standardized baseline uncertainty quantification framework to assess and compare accuracy. Adopting an algorithmic crediting approach with standardized uncertainty quantification would enable a virtuous cycle of iterative reductions in baseline uncertainty through intercomparisons, new data sources, and improved modeling methods. *Ex-post* project-level baselines may also fit into jurisdictional crediting approaches in order to attribute jurisdiction-scale reductions in forest loss to specific project activities [4].

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

Received 16 June 2023

(a) Project 1

(b) Project 2

(c) Project 3

(d) Project 4

(e) Project 5

(f) Project 6

(g) Project 7

**Figure 4: Scatter plots of observed vs. predicted (baseline) forest loss for placebos around each of the projects. Each color represents one year of data.**

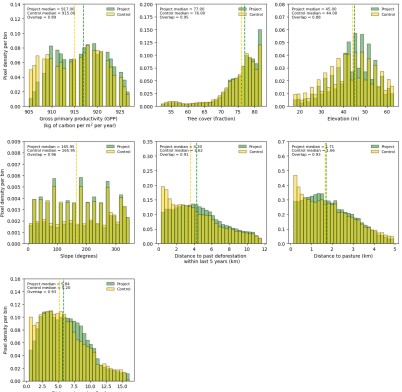

**Figure 5: Histograms of matching covariates distributed across the project (green) and control area (yellow) for an example project.**