# OpenReview forum: "Assessing forest carbon offset additionality with dynamic baselines and uncertainty quantification"
_KDD.org/2023/Workshop/Fragile_Earth — KDD 2023 Workshop Fragile Earth Submission_

### Official Review · Reviewer_1D5G · 2023-07-10
**This paper tackles the issue surrounding uncertainty quantification of the baselines used in the forest carbon projects. The problem statement is clearly communicated and methodology seems sound.  While details about the exhaustive K-NN algorithm could be presented in more detail, overall the paper is well rounded and addresses an important environmental issue, hence, I give it my acceptance.**

**Rating:** 7
**Confidence:** 2

**Review:**

-Summary : Authors address the issue of lack of uncertainty quantification used in the baselines in forest modeling which has potential impact in misappropriating carbon credits. Statistical Machine learning technique of Exhaustive K-NN has been employed
- Strengths:
      - a. Using ML, the paper tackles an important climate related issue sufficiently well.
      - b. Experimental evaluation has been conducted with metrics to evaluate the methodology.
      - c. Both current methodologies and potential future work are thoroughly addressed.


- Weaknesses:
      - a. Details on Exhaustive K-NN algorithm are not presented.
      - b. Even though limitations of current methods has been presented , a quantitative comparison with other methods in the literature
                 is lacking.

---

### Official Review · Reviewer_CDkH · 2023-07-13
**Review of "Assessing forest carbon offset additionality with dynamic baselines and uncertainty quantification"**

**Rating:** 7
**Confidence:** 3

**Review:**

In this paper, the authors introduce a dynamic baseline model based on remote sensing data and k-NN matching and apply a novel uncertainty quantification framework to assess the accuracy of dynamic baseline model. Using their proposed metric, the authors show the over/under-crediting relative to a project developer baseline for the seven REDD+ projects.

---

### Official Review · Reviewer_PfLq · 2023-07-16
**Review of "Assessing forest carbon offset additionality with dynamic baselines and uncertainty quantification"**

**Rating:** 7
**Confidence:** 4

**Review:**

Summary:

The authors introduce a dynamic baseline model based on remote sensing data and nearest neighbors matching and apply a novel
uncertainty quantification framework to assess the accuracy of this model. The case study in Brazil shows that several instances of consistent over or under-estimation of emissions reductions, suggesting potential inaccuracies in current carbon offset measurements which highlights the importance of our dynamic baseline and uncertainty quantification in enhancing the effectiveness of REDD+ and similar forest carbon projects.

Strengths:
- The paper is well written.
- The proposed pixel-level nearest neighbors-based approach is interesting and technically sound.

Weaknesses:
- In uncertainty quantification, can the authors explain more why the Project 2 has high RMSE and bias?

---

### Decision · Program_Chairs · 2023-07-19

**Decision:**

Accept (Oral)

**Comment:**

Congratulations!

We are pleased to inform you that your submission: Assessing forest carbon offset additionality with dynamic baselines and uncertainty quantification has been accepted to The KDD 2023 Workshop Fragile Earth: AI for Climate Sustainability - from Wildfire Disaster Management to Public Health and Beyond.

Camera ready deadline is ** July 24 AOE **.  Please log in to OpenReview and prepare your camera-ready version based on the reviews. Formatting rules are the same as for the initial submission and submissions must adhere to KDD 2023 guidelines available at https://authors.acm.org/proceedings/production-information/taps-production-workflow.

Again, congratulations on the acceptance of your paper!  We look forward to seeing you at the workshop on Aug 7, 2023.

The Fragile Earth Workshop Proceeding Chairs